# *ALPK1* hotspot mutation as a driver of human spiradenoma and spiradenocarcinoma

Mamunur Rashid[1,14], Michiel van der Horst[2,14], Thomas Mentzel[3], Francesca Butera[4], Ingrid Ferreira[1], Alena Pance [1], Arno Rütten[3], Bostjan Luzar[5], Zlatko Marusic[6], Nicolas de Saint Aubain[7], Jennifer S. Ko[8], Steven D. Billings[8], Sofia Chen[1], Marie Abi Daoud[9], James Hewinson[1], Sandra Louzada[1], Paul W. Harms[10], Guia Cerretelli[11], Carla Daniela Robles-Espinoza[1,12], Rajiv M. Patel[10], Louise van der Weyden[1], Chris Bakal [4], Jason L. Hornick[13], Mark J. Arends [11], Thomas Brenn[9,11,15] & David J. Adams[1,15]

Spiradenoma and cylindroma are distinctive skin adnexal tumors with sweat gland differentiation and potential for malignant transformation and aggressive behaviour. We present the genomic analysis of 75 samples from 57 representative patients including 15 cylindromas, 17 spiradenomas, 2 cylindroma–spiradenoma hybrid tumors, and 24 low- and high-grade spiradenocarcinoma cases, together with morphologically benign precursor regions of these cancers. We reveal somatic or germline alterations of the *CYLD* gene in 15/15 cylindromas and 5/17 spiradenomas, yet only 2/24 spiradenocarcinomas. Notably, we find a recurrent missense mutation in the kinase domain of the *ALPK1* gene in spiradenomas and spiradenocarcinomas, which is mutually exclusive from mutation of *CYLD* and can activate the NF-κB pathway in reporter assays. In addition, we show that high-grade spiradenocarcinomas carry loss-of-function *TP53* mutations, while cylindromas may have disruptive mutations in *DNMT3A*. Thus, we reveal the genomic landscape of adnexal tumors and therapeutic targets.

[1] Experimental Cancer Genetics, Wellcome Trust Sanger Institute, Cambridge CB10 1SA, UK. [2] Department of Pathology, Maasstad Hospital, Maasstadweg 21, Rotterdam 3079 DZ, The Netherlands. [3] Dermatopathologie Friedrichshafen, Siemensstrasse 6/1, 88048 Friedrichshafen, Germany. [4] Dynamical Cell Systems Laboratory. Chester Beatty Laboratories, Division of Cancer Biology. Institute of Cancer Research, London SW3 6JB, UK. [5] Institute of Pathology, Medical Faculty University of Ljubljana, Korytkova 2, Ljubljana 1000, Slovenia. [6] University Hospital Center Zagreb, Kispaticeva 12, 10 000 Zagreb, Croatia. [7] Institut Jules Bordet, 121-125, boulevard de Waterloo, 1000 Brussels, Belgium. [8] Department of Pathology, Cleveland Clinic, Cleveland, OH 44195, USA. [9] Departments of Pathology & Laboratory Medicine and Medicine and The Arnie Charbonneau Cancer Institute, Cumming School of Medicine, University of Calgary, Calgary, AB T2L 2K8, Canada. [10] Departments of Pathology and Dermatology, University of Michigan Medical School, 2800 Plymouth Road, Ann Arbor, MI 48109-5602, USA. [11] Division of Pathology, Cancer Research UK Edinburgh Centre, The University of Edinburgh, Institute of Genetics & Molecular Medicine, Crewe Road, Edinburgh EH4 2XR, UK. [12] Laboratorio Internacional de Investigación sobre el Genoma Humano, Universidad Nacional Autónoma de México, Campus Juriquilla, Blvd Juriquilla 3001, Santiago de Querétaro 76230, Mexico. [13] Department of Pathology, Brigham and Women's Hospital, Harvard Medical School, Boston, MA 02115, USA. [14]These authors contributed equally: Mamunur Rashid, Michiel van der Horst. [15]These authors jointly supervised the work: Thomas Brenn, David J. Adams. Correspondence and requests for materials should be addressed to D.J.A. (email: da1@sanger.ac.uk)

There are two main groups of adnexal tumors: those that are associated with the female reproductive system, including tissues such as the ovaries, fallopian tubes, and the connective tissues that surround these structures, and skin adnexal tumors of cutaneous origin. The word "adnexa" is Latin in origin and refers to the appendages of an organ. Spiradenoma and cylindroma are closely related benign skin adnexal tumors with sweat gland differentiation. They show histological similarities and may represent part of a morphological spectrum, further evidenced by rare spiradenoma–cylindroma hybrid tumors. The majority of tumors are sporadic and present as solitary nodules. Spiradenomas show a predilection for the extremities, while cylindromas commonly occur on the head and neck[1]. Occasionally, they may be multiple in the setting of the Brooke–Spiegler syndrome (BSS), a rare autosomal-dominant inherited disorder characterized by cylindromas, spiradenomas, and/or trichoepitheliomas in individuals with germline mutation of the *CYLD* gene[2]. Malignant transformation in spiradenoma (spiradenocarcinoma) and, less frequently, cylindroma (cylindrocarcinoma) is a rare event. Histologically, these tumors are composed of a benign precursor and a morphologically distinct malignant component, which may be further subdivided into low grade or high grade[3]. The morphology of these tumors appears to be a good predictor of outcome. Morphologically low-grade tumors have potential for local recurrence, while disseminated disease and disease-related mortality is largely limited to high-grade carcinomas[3–6]. Little is known about the underlying genetic events that drive these tumors. Cylindromas are characterized by mutations in *CYLD* and approximately two-thirds of sporadic cylindromas have also been reported to carry the *MYB-NFIB* fusion gene, which leads to overexpression of MYB, analogous to adenoid cystic carcinoma[7–9]. No genetic data are available for spiradenomas, and the events leading to malignant transformation and to the more aggressive behavior of the high-grade tumors are largely unknown. As yet, only mutations in the *TP53* gene have been reported in the malignant tumors[10,11].

To improve understanding of these rare diseases, we perform a comprehensive genomic characterization of samples from a large collection of representative patients and detail the driver gene landscape and biological processes that are operative. Notably, we find a hotspot driver mutation in *ALPK1* that defines spiradenoma and spiradenocarcinoma cases.

## Results

**Sample ascertainment and whole-exome sequencing.** Samples were obtained through the University of Edinburgh Tissue Bank with ethical approval obtained under REC 15/ES/0094. Analysis of these samples was also approved by the Sanger Human Materials and Data Management Committee (HMDMC). Cases were independently reviewed by two dermatopathologists to confirm diagnoses. In total, 75 samples underwent next-generation sequencing, 52 with paired adjacent normal/germline DNA (from 42 patients), while the remaining 23 samples (15 patients) without matched normal/germline DNA were used as a validation cohort (Supplementary Data 1). Capillary sequencing was also performed on 10 cases from 10 additional patients to validate a hotspot mutation as described below. A full breakdown of the samples used at the various stages of analysis and the available clinical characteristics of each patient is provided in Supplementary Data 1. Briefly, high- and low-grade spiradenocarcinoma, benign spiradenoma, and dermal cylindroma patients had a median age of 72.5, 61.5, 58, and 60 years at diagnosis, respectively. Notably, four patients (one cylindroma, one spiradenoma, one patient with a high-grade spiradenocarcinoma, and a patient with both a cylindroma–spiradenoma hybrid hybrid tumor and a

high-grade spiradenocarcinoma) were previously diagnosed with Brooke–Spiegler syndrome. Half of the tumors (37/68; 54%) were located on the head and neck area, while the remaining cases were from the trunk (19/68; 28%) or extremities (7/68; 10%). The tissue sites for the remaining 8% of tumors (5/68) were unknown. Formalin-fixed paraffin-embedded (FFPE) cores were collected from each tumor and DNA extracted, while uninvolved adjacent skin (epidermis/dermis/superficial subcutis) was used to obtain normal/germline DNA where available (referred to here as adjacent normal/germline). For several spiradenocarcinomas, we analyzed both low- and high-grade regions (Supplementary Data 1). DNA samples were whole-exome sequenced on the Agilent/Illumina platform at the Wellcome Trust Sanger Institute, generating a median depth of 60× coverage (after duplicate removal and read clipping).

**The somatic mutational landscape of adnexal tumors.** DNA-sequencing data from the 52 tumor/germline pairs were subjected to somatic variant calling (see the Methods section), resulting in the identification of 1124 somatic point mutations in exons, of which 817 were protein altering and 307 were silent mutations. The number of somatic single-nucleotide variants (SNVs) varied markedly between individual tumor samples (mean 21.6 mutations, range 2–144) (Fig. 1, Supplementary Data 2). In addition to SNVs, we also called 219 small insertion/deletions (indels) (Supplementary Data 2). Recurrently altered cancer driver genes included *CYLD* (15 cases [14 cases with protein coding/essential splice site changes, 1 case with a splice region change]), *NRAS* (p. Q129E, p.Q61K in the same sample), *AKT1* (p.E17K in three cases), *TP53* (p.E286K, p.G266E, and p.R248Q and indels p. D228Fs*20, p.R209Fs*6), and *DNMT3A* (p.R556M, p.R320*, E213_splice, E585_splice) (Fig. 1). All mutations shown were validated using high-depth (median depth of coverage 117×) targeted exome capture across all samples where DNA was available (Agilent design ID: S3065404) (Supplementary Data 1). To further validate our variant calls and to determine the accuracy of our whole-exome sequence capture analysis, we used our targeted exome data to assay a further 119 randomly selected somatic variants revealing an overall validation rate of 82%. For indels the validation rate was 73%. A pan-cancer analysis revealed that in comparison with cancers sequenced by The Cancer Genome Atlas (TCGA), the tumors sequenced here have a low somatic point mutation burden in the exome and fall within the range of 0.04–2.88 mutations/Mb (Supplementary Fig. 1), a frequency similar to thyroid cancer and uveal melanoma. Generally, cylindromas were found to carry more mutations than the other tumor types (Wilcoxon test $P = 0.0153$). Potential associations between the number of somatic mutations and age, sex of the patient, and tumor site were examined using a generalized linear model. No significant relationships with any individual clinical feature were observed. An overview of the genomic landscape, including all available clinical characteristics for these cases, can be found in Supplementary Fig. 2.

**Identification of driver genes in adnexal tumors.** A typical tumor cell may contain tens to thousands of somatic mutations distributed across hundreds of genes. Only a handful of these genes when mutated confer a selective growth advantage and thus may facilitate the promotion of tumor growth[12]. We applied two independent driver gene discovery tools: IntOGen and dNdScv to detect potential driver genes in our adnexal tumor cohort[13,14]. The IntOGen driver gene prioritization framework combines scores from SIFT, PolyPhen2 (PPH2), and MutationAssessor (MA), to calculate the functional impact bias (FM bias) of mutations in genes against a background distribution[13,15–17].

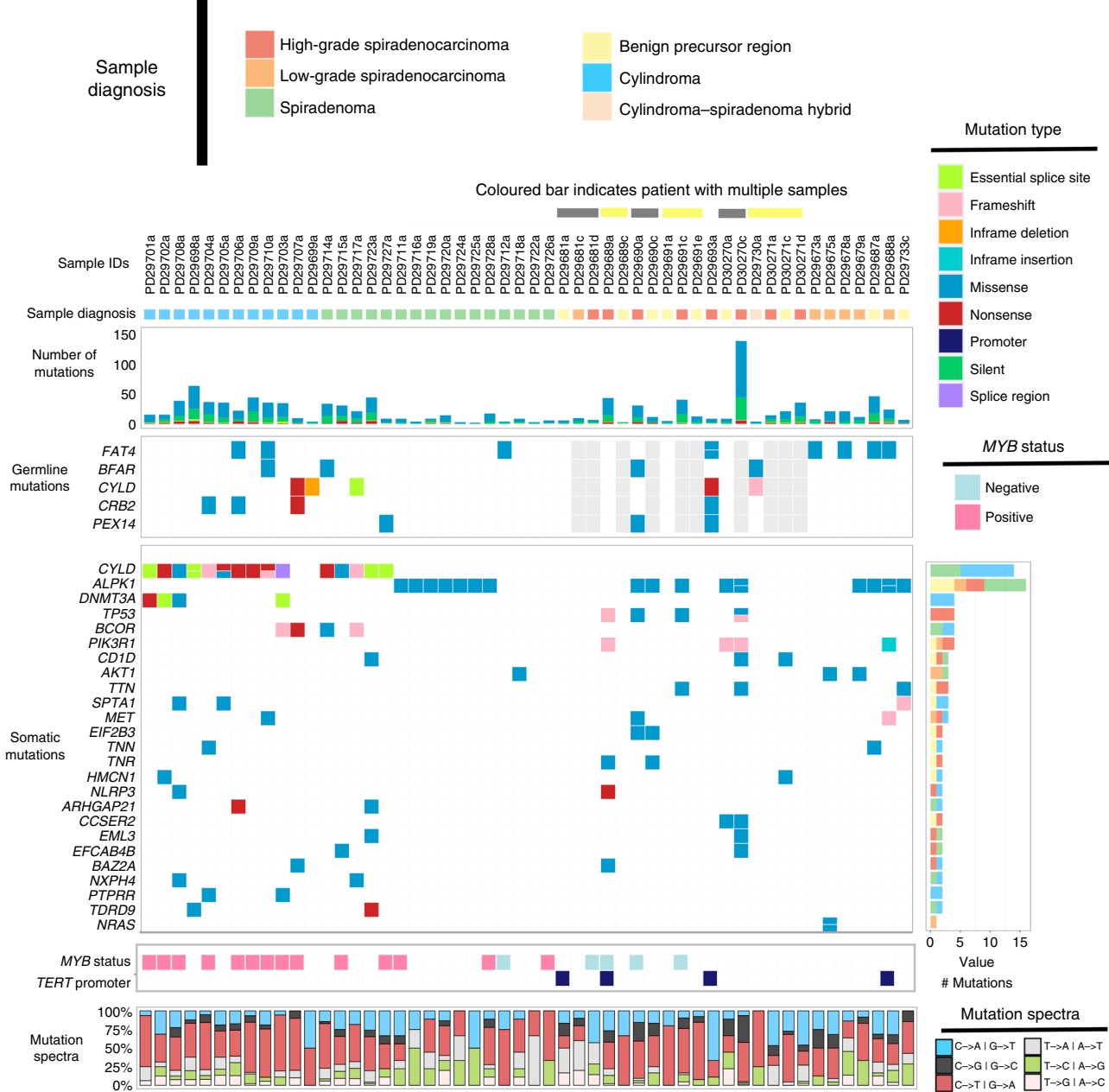

**Fig. 1** The driver gene landscape of skin adnexal tumors. Genomic data for the 52 cases where matched tumor/normal DNA-sequencing data were available. Additional cases are shown in Supplementary Fig. 5. The germline and somatic mutations in this plot were validated by high-depth targeted exome sequencing. Only mutations in coding regions are shown, except for *TERT* promoter variants and the splice region mutation in *CYLD*. Note that PD30271 and PD29730 are the same patient who had multiple tumors analyzed

Using this approach, genes computed to have a significant functional impact score (OncodriverFM q value) are reported as drivers. dNdScv on the other hand is a maximum likelihood-based method used to quantify positive selection of genes mutated in cancer, using the ratios of missense and disruptive mutations vs. synonymous mutations. We performed driver gene analyses using both of the aforementioned workflows, using somatic mutations from the cylindromas, spiradenomas, and high-grade and low-grade spiradenocarcinomas. A consensus of these two approaches is reported here. *CYLD* and *DNMT3A* were identified as statistically significant driver genes in cylindroma. *CYLD* was also reported as a driver gene in spiradenoma, while the tumor suppressor gene *TP53* was found to be significantly enriched with mutations in high-grade spiradenocarcinoma.

Notably, *ALPK1* was recurrently mutated at a hotspot position and was reported as a driver event in both spiradenoma (both methods) and spiradenocarcinoma (only by IntOGen), and is discussed in detail below. This mutation was absent from cylindromas. A complete list of the driver genes and significance values for each adnexal tumor type can be found in Supplementary Data 3. This table also provides the aggregate number of missense, nonsense, and synonymous mutations identified in each gene.

**A recurrent *ALPK1* mutation in spiradenoma and spiradenocarcinoma.** The *ALPK1* (α-kinase 1) gene is a member of the α-kinase family and is located on chromosome 4q25[18]. Recent studies have indicated that the expression of *ALPK1* during

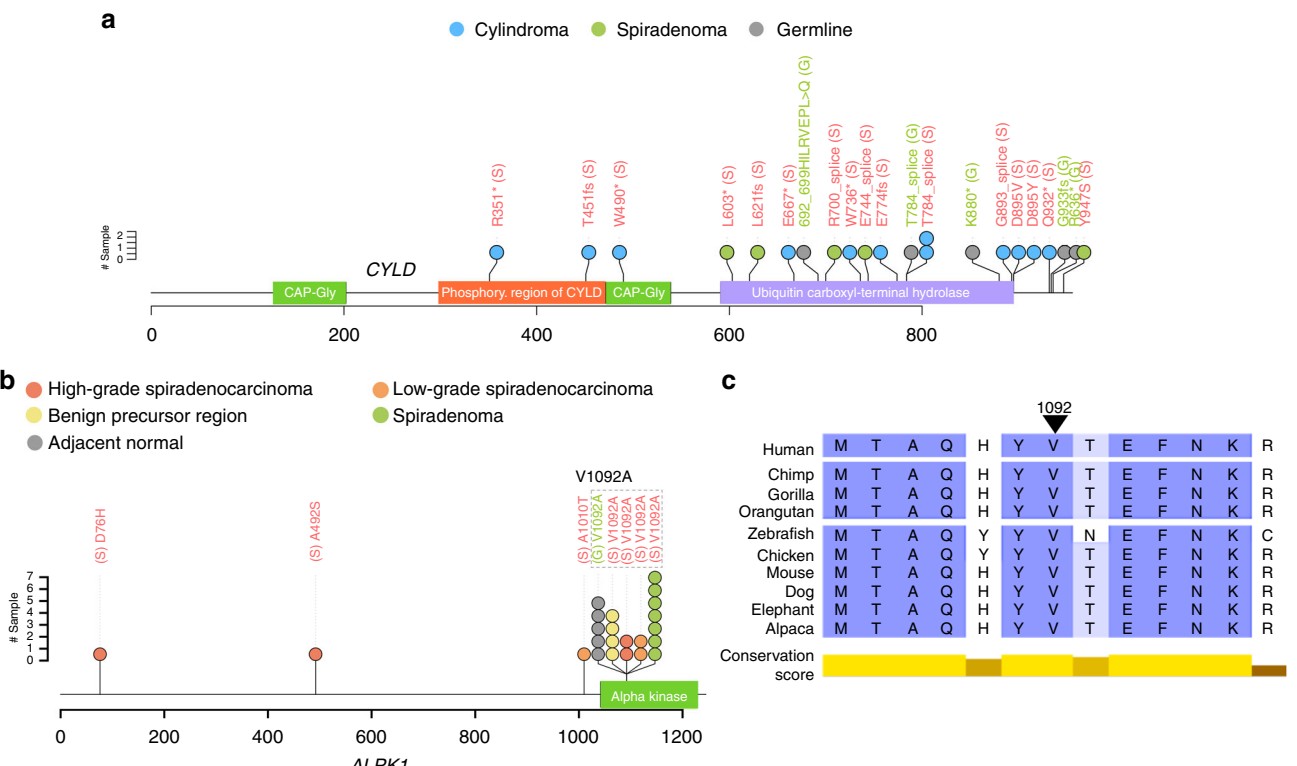

**Fig. 2** Mutations identified in *CYLD* and *ALPK1*. Variants in *CYLD* (**a**) and *ALPK1* (**b**) against the translation of the longest transcript of these genes (ENST00000458497.5 and ENST00000311559.13). Protein domains are from UniProt. All of the variants shown were validated by high-depth targeted exome sequencing. Adjacent normal represents morphologically normal tissue from the same block as the tumor which was used as a germline sample for somatic variant calling. Variants in red were called somatically. Variants in green were called from the adjacent normal tissue. The color of the circles indicates tumor/tissue type. The somatic splice region mutation in PD29703a in *CLYD* is not shown. **c** Protein alignment of ALPK1 across vertebrates. The conservation score represents constrained elements in multiple alignments by quantifying substitution deficits. The arrow indicates the position of the p. V1092 residue in humans

infection/inflammation can result in the activation of nuclear factor-kappa-B (NF-κB) signaling[19,20]. Somatic mutation of *ALPK1* in 32/1397 lung cancer samples (2.29%) and 29/781 colorectal cancer samples (3.71%) has recently been reported[18], and *ALPK1* has been shown to function as an oncogene in oral squamous cancers[21].

We discovered a recurrent somatic hotspot mutation in the alpha-kinase domain of the *ALPK1* gene (p.V1092A) in 7/16 spiradenomas, 2/8 high-grade, and 2/6 low-grade spiradenocarcinomas (Figs. 1 and 2, Supplementary Data 2) from our discovery cohort. All mutations were validated using targeted gene panel sequencing (see Methods). The hotspot mutation (p. V1092A) was also validated via Sanger sequencing in 8/11 of the aforementioned cases identified by whole-exome sequencing. The position was also tested in seven wild-type samples, all of which were confirmed to be mutation negative (Supplementary Data 1). Interestingly, in several cases (*n* = 5), we observed the *ALPK1* p. V1092A mutation in the adjacent morphologically normal tissue (in addition to the tumor), from which the normal/germline DNA for somatic variant calling was extracted. The average mutant allele fraction of the mutation in these samples was 0.32, suggesting that they are clonal or present in a significant proportion of cells. None of the other somatic mutations in the corresponding tumor sample were found in the sequence data from the adjacent morphologically normal tissue, making extensive tumor-to-normal contamination unlikely (see Methods). This mutation was also observed in sequence data generated from benign precursor regions (*n* = 4), suggesting that the *ALPK1*

p.V1092A mutation may be an early founder/truncal mutation, or is associated with a field change, as has been widely reported for other cancers, particularly skin[22]. Interestingly, mutation of *ALPK1* was mutually exclusive (*q*-value 0.00146[23]) from mutation of *CYLD*[22,24] (Fig. 1). To further confirm the presence of the *ALPK1* p.V1092A mutation, a further 10 spiradenoma tumor/normal pairs were tested via Sanger sequencing and the p. V1092A mutation was observed in six tumors.

**Mutation of *CYLD* in adnexal patients and tumors**. *CYLD* (CYLD lysine 63 deubiquitinase) encodes a cytoplasmic protein with three cytoskeletal-associated protein–glycine-conserved (CAP–GLY) domains and functions as a deubiquitinating enzyme and tumor suppressor[25]. CYLD regulates the NF-κB pathway, which plays important roles in cell growth and survival[26,27]. Germline mutation of *CYLD* is associated with Brooke–Spiegler syndrome, which may present with cylindroma, cylindromatosis, trichoepithelioma, and/or spiradenoma[2]. Eleven of the twelve cylindroma patients we sequenced carried either germline or somatic protein-altering mutations of *CYLD*. The final cylindroma case (PD29703a) was found to carry a somatic splice region mutation (16: 50815325 A/G) located three bases away from the splice junction. *CYLD* mutations were also found in 31% (5/16) of the spiradenomas (Fig. 1). All four patients with a prior Brooke–Spiegler syndrome diagnosis, whose germline we sequenced, carried a germline *CYLD* mutation (Supplementary Data 1). The protein-altering mutations in *CYLD* are shown in Fig. 2a.

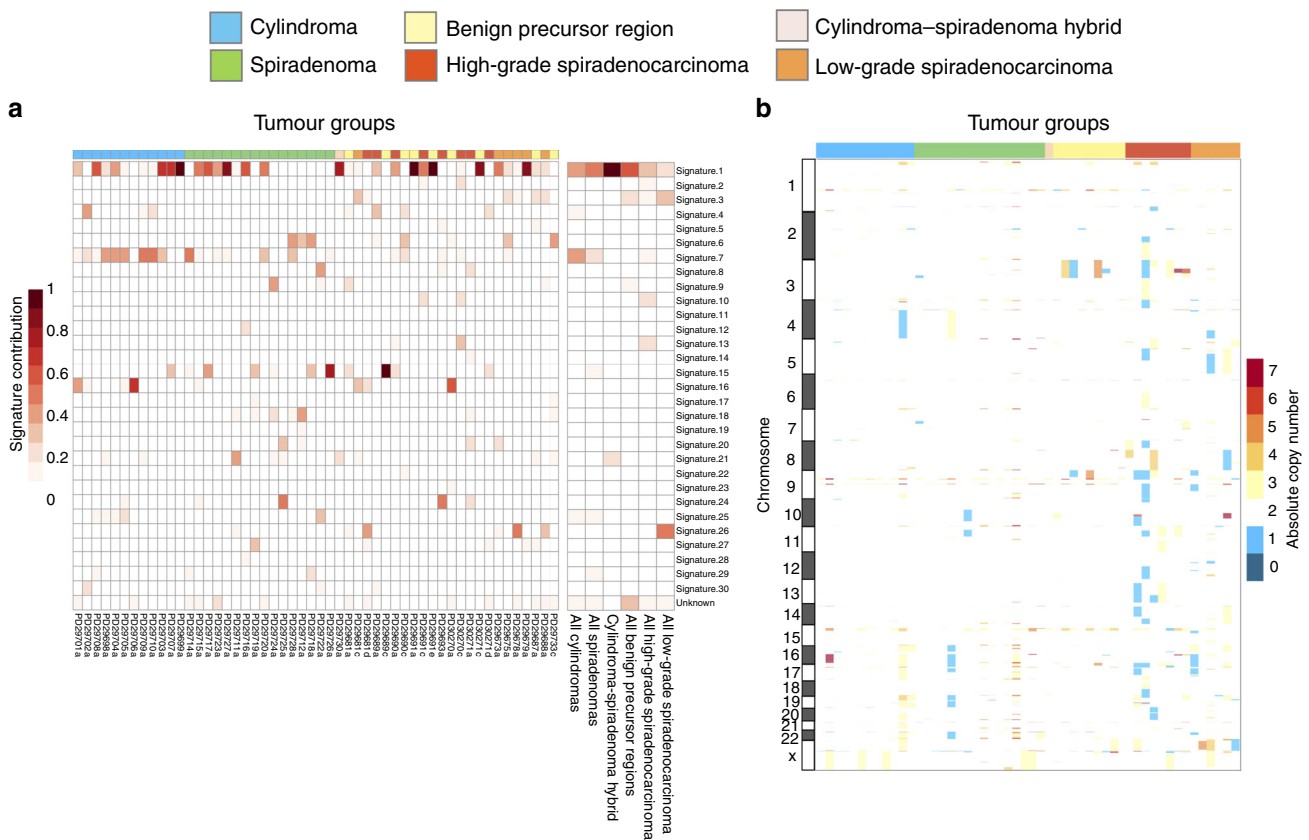

**Fig. 3** The somatic genetic landscape of adnexal tumors. **a** The contribution of published mutational signatures in adnexal tumors was computed using deconstructSigs[32]. Total contribution per sample adds up to one. For this analysis, we used all variants, including those in noncoding regions, such as 5′ and 3′ UTRs. **b** The copy-number landscape of adnexal tumors. This analysis was performed using Sequenza to define the absolute copy number for chromosomal segments. These analyses were performed using the tumors shown in Fig. 1

**Promoter and regulatory mutations**. *Cis*-regulatory elements control the transcription of genes, and mutations in these regions can potentially lead to aberrant protein production and tumorigenesis. Exome sequencing is not well equipped to detect *cis*-regulatory element mutations, as it is designed to capture protein-coding regions. However, sufficient coverage (>10× read coverage) around exon boundaries allowed us to investigate the status of proximal regulatory elements, such as promoters. Detected noncoding mutations were scored for pathogenicity, weighting them with a CADD (Combined Annotation-Dependent Depletion) variant deleteriousness score[28] (see Methods). Mutations were also annotated in the regulatory regions of known cancer driver genes. In this way, we identified mutations in the *TERT* promoter region (C228T and C250T) in four spiradenocarcinomas, known hotspot positions in other cancers[29]. Somatic mutations in the proximal regulatory regions of other genes, such as *SPTA1, HMCN1,* and *COL11A1*, were also detected (Supplementary Data 2).

**Mutational processes in adnexal tumors**. Somatic mutations in tumor cells may be the consequence of aberrant endogenous processes, such as defective DNA repair, or due to exogenous factors, such as exposure to carcinogens. The imprint of a mutational process on the DNA sequence of the genome is commonly referred to as a mutational signature[30]. Analysis of mutational signatures has led to a better understanding of the underlying biological processes associated with a number of cancers and has also allowed patient stratification for therapy[31].

To assess the presence of published human cancer mutational signatures in the catalogue of somatic mutations from adnexal tumors, we used deconstructSigs (see Methods)[32]. This approach computes the weighted contribution of the 30 published COSMIC signatures and one additional unknown signature to the collection of somatic variants called from each sample. The heatmap in Fig. 3a represents the contribution of these signatures across all adnexal tumor subtypes. In more than a quarter (26.92%) of tumors, the contribution of signature 1 was greater than 0.5, meaning most mutations in these samples can be attributed to this signature. Signature 1 is an endogenous mutational process associated with spontaneous deamination of 5-methylcytosine, which is often correlated with age[33]. Cylindromas were also enriched for signature 7, which is predominantly found in skin cancers as a result of ultraviolet (UV) light exposure. The predilection of cylindromas to form on the head and neck is likely to explain this signal. We also performed an analysis combining mutations for each tumor type together and again identified signature 1 and signature 7 in cylindromas, while several low-grade spiradenocarcinomas showed a signal for signature 26, which is thought to be associated with DNA mismatch repair[30].

**Somatic DNA copy-number alterations**. The copy-number status of our adnexal samples was assessed using Sequenza, an allele-specific copy-number analysis algorithm that uses matched tumor-normal pairs[34]. Sequenza reported a total of 1577 somatic copy-number changes (1350 gains and 227 losses) across 52 tumors. Several high-grade spiradenocarcinomas showed large copy-number changes, while low-grade spiradenocarcinomas demonstrated a comparably lower number of copy-number events, although a larger number of samples will be required to

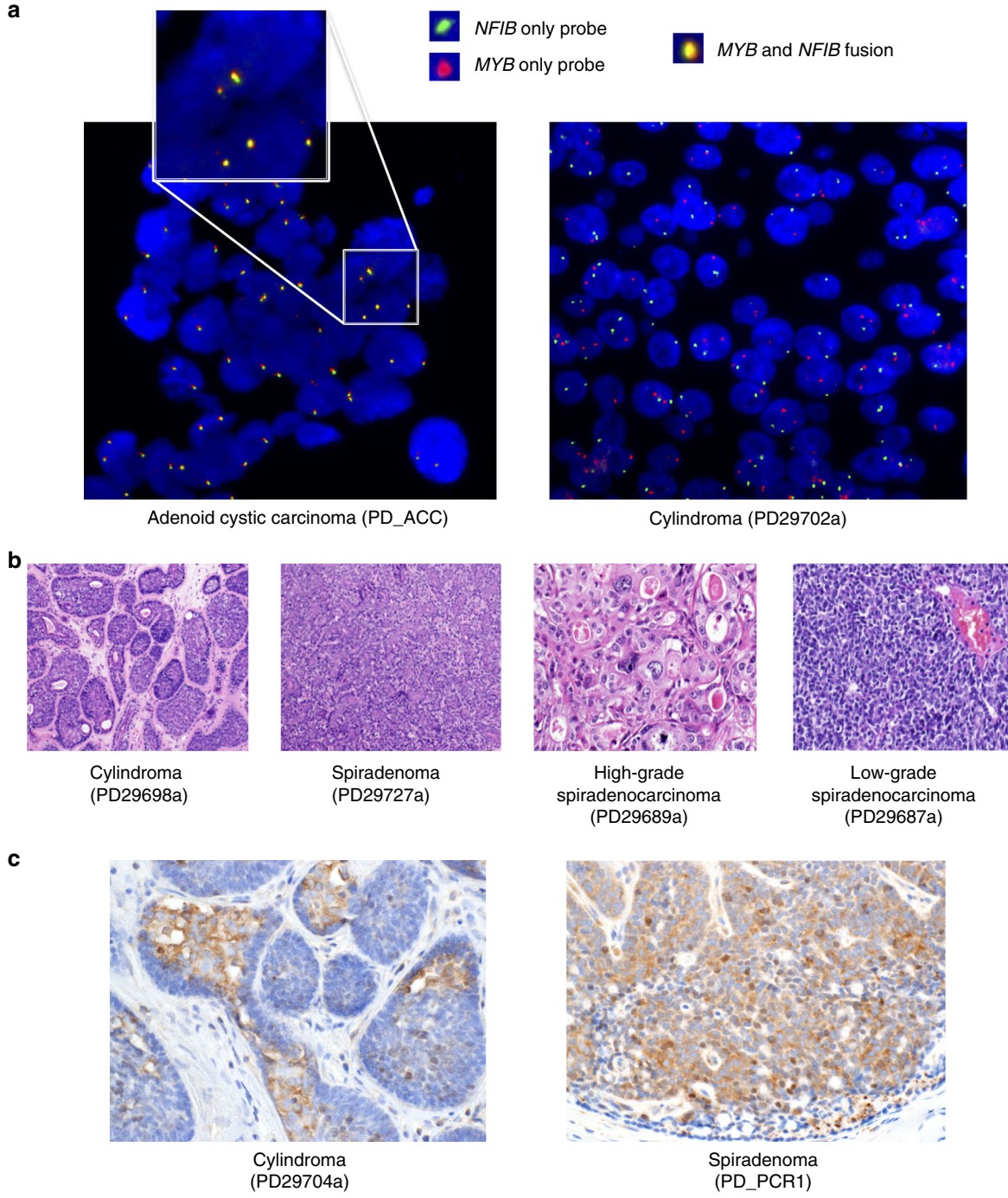

**Fig. 4** Assessment of the *MYB-NFIB* fusion and p65 expression in adnexal tumors. **a** Fluorescence in situ hybridization (FISH) imaging of the *MYB-NFIB* fusion in an adenoid cystic carcinoma and assessment in cylindroma samples. Previous reports have suggested that adnexal tumors, such as cylindomas, carry *MYB-NFIB* fusions which have been associated with *MYB* overexpression[8]. The left panel shows an adenoid cystic carcinoma, which is a positive control for the fusion event. Yellow signal results from the overlap of the green *NFIB* probe and red *MYB* probe. Right panel: a representative cylindroma which was fusion negative. **b** Representative histopathological images of a cylindroma at ×100 magnification, spiradenoma at ×100 magnification, high-grade spiradenocarcinoma at ×200 magnification, and a low-grade spiradenocarcinoma at ×200 magnification. **c** p65 immunohistochemistry of a *CYLD* mutant cylindroma (left) and an *ALPK1* p.V1092A mutant spiradenoma (right) at ×20 magnification

fully explore the copy-number landscape of these tumors. Cylindromas and spiradenomas generally showed few copy-number changes, as did morphologically benign precursor regions. Genome-wide copy-number profiles across all subtypes are reported in Fig. 3b.

**The *MYB-NFIB* fusion in adnexal tumorigenesis.** A previous report has suggested a role for *MYB-NFIB* fusions in the pathogenesis of both adenoid cystic carcinoma and cylindroma[8,9]. Using dual-color FISH, we analyzed 21 cases, including 13 cylindromas, 7 spiradenomas, and 1 cylindroma–spiradenoma hybrid tumor in addition to an adenoid cystic carcinoma case known to carry the *MYB-NFIB* fusion as a control. This analysis revealed that, despite previous reports, none of the cylindromas were found to carry the fusion event[8]. The *MYB-NFIB* fusion was also absent from the spiradenomas and cylindroma–spiradenoma

hybrid tumor (Fig. 4a, b, Supplementary Figs. 3 and 4). Over-expression of MYB was, however, confirmed in cylindroma and spiradenoma cases using immunohistochemistry (Fig. 1 and Supplementary Data 4), suggesting that other mechanisms of gene overexpression are operative.

**Germline analysis of adnexal tumor patients**. As mentioned above, we identified germline *CYLD* mutations in all four patients previously diagnosed with Brooke–Spiegler syndrome. A germline *CYLD* mutation (in-frame deletion) was also detected in an additional patient with no prior Brooke–Spiegler syndrome diagnosis (PD29699) (Fig. 1). To extend the analysis of germline variation in patients from our cohort, we used samtools mpileup and the bcftools variant genotyping strategy[35]. We assessed the coding mutation burden per gene using a Fisher's exact test (see Methods) (Supplementary Data 2)[36] using variants from the 42 cases where adjacent normal/germline exome sequence had been generated. From this analysis, *CYLD* was found to carry significantly more mutations than expected (Benjamini–Hochberg (BH)-adjusted $p$-value 0.01), reconfirming its well-established role as an adnexal tumor predisposition gene. We also detected a significantly high number of mutations in *FAT4, BFAR, CRB2,* and *PEX14* (BH-adjusted $p$-value < 0.05). *FAT4* is a member of human *FAT* gene family which encodes a large transmembrane protein consisting of multiple extracellular cadherin domains and a cytoplasmic domain that can interact with signaling molecules[37]. This gene is homologous to fat in *Drosophila*, a known tumor suppressor gene[38]. It should be noted, however, that *FAT4* has been reported as "disease-associated" in several studies, which might suggest a high rate of polymorphism[39–41]. *BFAR*, the bifunctional apoptosis regulator, plays a role in the regulation of cell death and in this way could contribute to tumorigenesis[42]. Notably for *FAT4, BFAR, CRB2,* and *PEX14,* we did not identify somatic mutations in the wild-type allele of these genes in the cases carrying germline mutations, suggesting that if they are contributing to tumor formation, they most likely do not function as classical tumor suppressors. Further, samples with germline variants in these genes also had germline or somatic loss-of-function alleles of *CYLD*, making these less likely candidate predisposition genes. We next asked if mutations of known pathogenicity were found in the germline of any of our adnexal cases. In this way, we found 14 pathogenic or likely pathogenic variants, including missense mutations in *PTEN* and *NSD1* (Supplementary Data 2) (ClinVar database (dbSNP build 144)). The *PTEN* mutation we found has been reported in a single individual in ClinVar with a Cowden-like syndrome and changes a phenylalanine to a serine (P200S). This substitution occurs at a position that is conserved across species. In silico analysis predicts that this variant is probably damaging to protein structure/function. Missense (disease-causing) mutations in nearby residues (V191G, M198K, T202I, M205V, and S207R) have been reported, supporting the functional importance of this region of the protein. That said, the clinical records for our patient (PD29681) do not mention Cowden syndrome and the patient was 81 when their skin tumor was removed. Further, we did not detect a somatic mutation in the other allele of *PTEN*. In contrast, our patient carrying a germline mutation in *NSD1* carries a well-established SOTO syndrome-associated allele (R2017W) predicted to be disruptive of SET domain function[43,44]. Thus, of the 42 adnexal patients analyzed here, we have shown that five patients carry germline mutations in *CYLD* and propose several other candidate genes as mediators of germline susceptibility for follow-up studies.

**Analysis of tumors without matched germline DNA**. For 52 of the samples in our cohort, we had matched tumor/adjacent normal-germline pairs (as described above). Matched germline

DNA was not available for a further 23 samples (15 patients; 3 cylindroma, 1 spiradenoma, 1 cylindroma–spiradenoma hybrid, 3 low-grade spiradenocarcinoma, and 7 high-grade spiradenocarcinoma), and thus we used the tumor sequences from these cases as a validation cohort to look for variants in genes identified from the above-mentioned analyses. We first called variants against an unmatched normal sample (Supplementary Data 1) and then filtered these data using variants in the ExAC database[36] (with an allele frequency > 0.0001) and with variants from an in-house panel of 100 normal-germline exomes. We next focused on genes identified from our analysis of the discovery cohort (see Methods), revealing *ALPK1* p.V1092A mutations in one cylindroma–spiradenoma hybrid, two low-grade spiradenocarcinomas, and one spiradenoma. Loss-of-function mutations were also detected in *CYLD* in several cylindroma cases (PD29695a, PD29696a, and PD29700a) and in one low-grade spiradenocarcinoma (PD29676a). Tumors from two high-grade spiradenocarcinoma patients (PD29684 and PD29685) were found to carry frameshift deletions in *TP53* (p.P191fs*54 and p.T329fs*8). For each patient, the respective changes were present in all collected tumor samples, indicating that they maybe germline in origin or occur early in tumor development. An overview of the driver gene landscape and clinical characteristics of all 75 tumors/samples can be found in Supplementary Fig. 5.

**Functional studies of the *ALPK1* p.V1092A variant**. Given the role of *ALPK1* in the regulation of the NF-κB pathway in infection, we next asked if the *ALPK1* p.V1092A variant could activate NF-κB signaling and thus substitute for mutation of *CYLD*. To do this, we generated full-length *ALPK1* wild-type and p.V1092A mutant cDNA constructs in an expression vector. Since adnexal cell lines do not exist, we transfected these constructs into a panel of six cancer cell lines and assayed NF-κB activity by monitoring NF-κB nuclear localisation (Supplementary Fig. 6). Analysis in this way showed that the mutant construct increased NF-κB activity to a considerably higher level than the wild-type construct in MCF-7, WM266-4, and WM1552C cells, consistent with a role for this variant in driving tumor growth through the NF-κB pathway, akin to mutation of *CYLD*. To further confirm that the p.V1092A *ALPK1* mutation activates NF-κB signaling, we performed immunohistochemistry and found that p65 staining of *ALPK1* mutant spiradenomas was indistinguishable from staining of *CYLD* mutant cylindromas (Fig. 4c).

**Discussion**

The analysis of adnexal tumors in this study yielded several remarkable results. First, we identified a recurrent somatic missense *ALPK1* mutation (p.V1092A) in the kinase domain of this alpha-kinase and demonstrated that this mutation can activate NF-κB signaling in cell reporter systems. Importantly, *ALPK1* has previously been suggested to function as an oncogene[45]. Since kinases can be readily inhibited, this mutation represents a potential therapeutic target, which might be particularly advantageous in the advanced/metastatic setting, where effective treatments have not been identified. Second, we found driver genes not previously associated with adnexal tumors. For example, statistical analyses revealed significant enrichment of mutations in *DNMT3A* in cylindromas, a gene previously linked to hematopoietic malignancies, where it plays a role in the regulation of DNA methylation[46,47]. Further studies will be required to establish the direct functional role of these mutations in adnexal tumors and their effect on the epigenetic landscape. Mutations in genes such as *AKT1, BCOR,* and *PIK3R1* were also observed and these genes may also contribute to tumor development. In keeping with previous studies, we found frequent mutation of the

*CYLD* gene[7,48]. Somatic or germline *CYLD* mutations were found in 12/12 cylindroma patients (matched tumor-germline cases) with mutations also being observed in spiradenoma and high-grade spiradenocarcinoma cases. Notably, these mutations were mutually exclusive from the above-mentioned *ALPK1* variant. As the etiology of adnexal tumors is unknown, we performed a mutational signature analysis. This revealed not only the presence of signature 1 across all tumor types, which is age-associated, but also the UV-associated signature 7 in cylindromas, presumably because these tumors are generally found on the head and neck. There was also some suggestion of signature 26, associated with mismatch repair, in low-grade spiradenocarcinomas. Tumors in our adnexal collection were not only low in terms of their somatic mutation burden, but also appeared to lack significant copy-number alterations, the exception being several of the high-grade spiradenocarcinomas which, compared with other adnexal tumors, were replete with copy-number gains/losses. Finally, we identified germline variants in *CYLD* that have not been described previously, and thus represent new pathogenic alleles. We also found cases with pathogenic variants in the ClinVar database, including in *PTEN* and *NSD1*, suggesting potential adnexal tumor predisposition alleles. The identification of a patient with an *NSD1* mutation, which is associated with Sotos syndrome, is of particular interest since previous case reports suggest adnexal tumors in some patients with this condition[49]. These insights should be explored in a larger case series.

In summary, our paper reports the most comprehensive picture of the genomic landscape of adnexal tumors to date, including driver genes, copy-number alterations, and a potentially actionable kinase mutation and mutational signatures. We hope that these studies will help inform the management of patients with these malignancies.

## Methods

**Patients and samples.** Samples for whole-exome sequencing (WES) and targeted gene panel sequencing (TGPS) were collected from 57 patients and divided into a discovery (tumor/adjacent normal-germline pairs) and a validation cohort (tumor only). The discovery cohort contained 52 tumors/samples and matched adjacent normal/germline DNA from 42 patients. This cohort was used for the initial genomic profiling and driver gene analyses. Mutations from 23 additional samples (15 patients) from the validation cohort were also reported. From some lesions, we were able to obtain high-grade and low-grade spiradenocarcinoma regions which were sequenced and analyzed separately. A detailed description of each case/sample can be found in Supplementary Data 1. All diagnoses were confirmed by two independent pathologists. Ethical approval was obtained from the West Lothian Tissue bank. DNA was extracted using Qiagen kits.

**Whole-exome sequencing.** Exonic DNA was captured using the Agilent whole-exome capture kit (SureSelect All Exon V5). Captured material was indexed and sequenced on the Illumina Hiseq2500 platform at the Wellcome Sanger Institute to a median depth of 60×. Raw 75-bp pair-end sequencing reads were aligned with BWA (v0.7.12) to the GRCh37 human reference genome, producing a single Binary Alignment Map (BAM) file for each sample[50]. Duplicated reads resulting from PCR were marked with BioBamBam (v2.0.54)[35,50,51].

**Targeted gene panel resequencing.** To confirm our findings from whole-exome sequencing, we validated mutations in the top recurrently mutated genes using panel sequencing (Supplementary Data 5). Genomic regions for 550 genes were captured using Agilent custom pulldown baits. Captured material was indexed and sequenced on the Illumina Hiseq4000 platform to a median depth of 117×. Raw 75-bp pair-end sequencing reads were processed using the same pipeline, as that used for whole-exome sequencing described above.

**Somatic variant detection.** Somatic variants were detected using CaVEMan, an expectation maximization-based somatic substitution detection algorithm[52]. To ensure that tumor and normal were paired correctly for somatic variant calling and to avoid any possible sample swaps, we used genotype data from 20,000 randomly selected germline variants. A pairwise correlation coefficient between each sample pair is shown in Supplementary Fig. 7. Candidate somatic variants were then filtered for quality and to remove common population variants (ExAC allele frequency > 0.0001). Small insertion and deletion (indel) detection was performed using the cgpPindel pipeline (v0.2.4w)[53]. Detected indels were then filtered for

quality, sequence coverage in both tumor and normal, strand bias, and for overlap with known simple repeats or indels in an in-house normal panel.

**Variant quality control for FFPE artifacts.** Formalin fixation of tumor biopsies can have a detrimental impact on DNA integrity and introduce C > T/G > A sequencing artifacts[54]. These artifacts are more frequently observed at a 0.01–0.10 mutant allele fraction (MAF)[54]. To remove these variants, we used the following filters:

Tumor read depth (TRD) and adjacent normal/germline read depth (NRD) must be greater than or equal to 10.
Mutations with MAF < = 0.10 were kept only if the TRD and NRD were greater than or equal to 30.
Mutations with MAF < = 0.05 were kept only if TRD was greater than or equal to 100.

After filtering our somatic point mutations, the validation rate of variants from the whole-exome sequencing data was 82% for SNVs and 73% for indels, as determined by targeted sequencing (see above).

**Mutual exclusivity analysis of *ALPK1* and *CYLD* mutations.** Mutual exclusivity between *ALPK1* and *CYLD* was evaluated using the DISCOVER[23] co-occurrence and mutual exclusivity analysis tool. Somatic point mutations and small indels from all 52 tumors were combined into a single $N \times M$ binary data matrix, where each cell value $V_{i,j}$ ($i = 1 \ldots N$ [Number of genes], $j = 1 \ldots M$ [Number of tumors]) indicated the status of gene $i$ in tumor $j$. $V_{i,j} = 1$ if gene $i$ is mutated in tumor $j$ and 0 otherwise. Alteration status of all genes across all tumors was used to generate a null distribution for background alternation rate estimation. Finally, we computed pairwise mutual exclusivity between any two genes mutated in more than two tumors, taking the null distribution into account.

**Germline mutation burden analysis.** We applied an exome-wide Fisher's exact test to assess the significance of observing $n$ mutations in gene $X$ in our 42 germline samples, given gene $X$ has a mutation rate of $Y$ in a control population. To select an appropriate control population, we performed a principal component analysis using 2504 individuals across multiple populations from the 1000 Genomes Project phase3[55]. We randomly selected 2000 single-nucleotide polymorphic variants (SNPs) and to mitigate the impact of population-specific rare variants, we only selected SNPs with a population allele frequency of between 0.1 and 0.7. PCA analysis revealed that all 42 tumor–germline pairs were of European descent (Supplementary Fig. 8). Therefore, polymorphic variants from the ExAC database from individuals of non-Finnish European descent were used as a negative control. Readers should be mindful of the strengths and weakness of such an approach[56]. A Combined Annotation-Dependent Depletion (CADD) score filter was applied and only variants with a CADD score above or equal to 15 were taken forward for burden testing. We also ensured that only variants with sequence coverage of > 10× in both the case and control dataset were used. Finally, we applied a Fisher's exact test on every gene to estimate the likelihood of observing $n$ deleterious mutations, given the background mutation rate of that gene in the control population. The Benjamini–Hochberg method was used to correct for multiple testing and only genes with an adjusted $p$-value less than or equal to 0.05 were reported as significant.

**Mutational signature analysis.** To reduce the potential impact of artifacts from 5-methylcytosine deamination and degradation in our FFPE samples, low allelic fraction mutations (mutant allele fraction < 0.10 and read depth < 10) were removed from the signature delineation process (as outlined above). Somatic point mutations were then mapped to the 96 possible trinucleotide contexts, taking into account the probability of each mutation occurring in each trinucleotide within the human genome. We then applied deconstructSigs, a multiple linear regression-based algorithm to reconstruct the mutation profile of each tumor sample using a linear combination of predefined mutational signatures[32]. Thirty human cancer signatures as defined in Alexandrov et al., were used for the reconstruction and one "unknown" signature[30].

**DNA copy-number analysis.** To estimate allele-specific copy-number profiles, we used the Sequenza software package (v2.1.0), a probabilistic model-based algorithm applied to a segmented average depth ratio (tumor vs. normal) and B-allele frequency[34]. Preprocessing and analysis with Sequenza were performed as described in the Sequenza documentation, and fitted models were manually examined. For four tumor-normal pairs, a default fitted model suggested a very high ploidy. However, after manual inspection of the depth ratio and B-allele fraction data, an alternative solution closer to ploidy 2 was selected due to lack of evidence for high ploidy.

**Gene fusion analysis by fluorescence in situ hybridization.** Fusion gene analysis of the paraffin-embedded tissue sections was performed using the *MYB-NFIB* fusion/translocation FISH probe kit from CytoTest, following the manufacturer's protocol. The *MYB* 5′ probe covers the entire *MYB* gene along with upstream (5′)

and some downstream (3′) genomic sequences. The *NFIB* 3′ probe covers the 3′ (end) portion of the *NFIB* gene along with some adjacent genomic sequence. An adenoid cystic carcinoma (PD_ACC) case known to carry the fusion was used as a positive control.

**ALPK1 hotspot validation using Sanger sequencing.** DNA was extracted as above for exome sequencing. The region of interest of *ALPK1* was amplified using ThermoFisher Platinum HiFi *Taq* DNA polymerase (following the manufacturer's instructions) using the oligos shown below. Amplified products were sequenced by Sanger Sequencing (Eurofins) using the same oligos. Sequence traces were analyzed by visual inspection.

 *ALPK1* forward: 5′-TTGATCTCCTCTCTCTTACTCCA-3′
 *ALPK1* reverse: 5′-ATGCTAGCCTGATTATGTGGAA-3′

**Functional analysis of ALPK1 mutation by NF-κB reporter assays.** MCF-7, T-47D, WM266-4, WM1552C, PANC-1, and MIA PaCa-2 cells (obtained from the American Type Culture Collection [ATCC]) were seeded in T25 flasks to obtain 80% confluence for transfection. The following day, cells were transfected with 1.5 μg of WT or Mut *ALPK1* cDNA and 1.5 μg of RFP using the Effectene kit (Qiagen), according to the manufacturer's protocol. After 24 h, 4000 cells were transferred to six wells per line/construct in a 384-well TC-treated PelkinElmer Cell Carrier Ultra plate. After a further 24 h, cells were fixed with formaldehyde/PBS at a final concentration of 4% for 10 min at 37 °C. Cells were permeabilized in 0.2% Triton X-100/PBS (Sigma Aldrich) for 10 min and blocked in 2% BSA/PBS for 1 h at RT. Cells were stained with rabbit anti-p65/RELA NF-κB (Abcam; cat 16502; 1:500) for 2 h at RT and then Alexa 647 goat anti-rabbit IgG (Invitrogen, 1:500) for 1 h at RT. Cells were stained with 10 μg/ml Hoechst (Sigma Aldrich; cat 33258) for 10 min at RT. Images were taken using the PerkinElmer Opera confocal microscope and a 20× air objective. Image analysis was performed using custom image analysis scripts with PerkinElmer's Columbus 2.6.0 software.

**MYB expression by immunohistochemistry.** MYB overexpression in cylindromas has been reported in several earlier studies[57]. We attempted to assess the MYB expression status of 26 samples (11 cylindromas, 6 spiradenomas, and 9 high-grade spiradenocarcinomas) using immunohistochemistry (IHC) (Supplementary Data 4). IHC was performed on 4-μm-thick formalin-fixed paraffin-embedded whole-tissue sections following antigen retrieval with Target Retrieval solution (pH 6.1; Dako, Carpinteria, CA, USA) in a pressure cooker using a rabbit monoclonal anti-MYB monoclonal antibody (1:200 dilution; clone EP769Y; Abcam, Cambridge, MA, USA) and the Envision + polymer detection system (Dako). Immunohistochemistry for p65 was performed using an anti-NF-κB p65 antibody (1:5000 dilution; clone D14E12; Cell Signaling Technology, Danvers, MA, USA) as described for MYB above.

**Reporting summary.** Further information on research design is available in the Nature Research Reporting Summary linked to this article.

## Data availability

The data have been accessioned under the study EGAS00001001799 in the European Genome-phenome Archive. Source Data File 1 provides all of the variant calls in MAF and VCF format. All the other data supporting the findings of this study are available within the article and its supplementary information files and from the corresponding author upon reasonable request. A reporting summary for this article is available as a Supplementary Information file.

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

## Acknowledgements

This work was supported by grants from Cancer Research UK, the Wellcome Trust, and by the ERC Combat Cancer Project to DJA. C.B. is funded by the Stand Up to Cancer campaign for Cancer Research UK and by a Cancer Research UK Programme Foundation Award (C37275/A20146). We also wish to thank Dr. J.W.R. Meijer (Rijnstate Hospital, Arnhem, the Netherlands) on behalf of the PALGA group and Fengtang Yang for technical assistance.

## Author contributions

R.M., M.v.d.H., M.J.A., T..B and D.J.A. designed the study and oversaw the research. M. R., S.C., and C.D.R.E. performed data analysis. T.M., I.F., A.R., B.L., Z.M., N.d.S.A., J.K., S.D.B., M.A.D., P.W.H., R.M.P., J.L.H., M.J.A., and T.B. performed histopathological analysis of the tumor samples. F.B., A.P. and C.B. performed NF-κB assays. J.L.H. performed immunohistochemistry on tissue samples. L.v.d.W., J.H., and G.C. processed tissue samples for sequencing. S.L. performed FISH analysis. R.M., M.J.A., T.B., and D.J. A. wrote the paper will input from all other authors.

## Additional information

**Competing interests:** The authors declare no competing interests.

