## [Peer Review File · Nature Communications]

Reviewers' comments:

Reviewer #1 (Remarks to the Author):

Authors present the genomic analysis of Formalin-Fixed Paraffin-Embedded (FFPE) samples of skin adnexal tumors (cylindromas, spiradenomas, low- and high-grade spiradenocarcinoma as well as of morphologically benign precursor regions of these cancers. These are skin cancer with no or very little genomics information, so information presented by the study is important for a broad community of cancer researchers. The study indicated a specific recurrent somatic missense mutation in the kinase domain of ALPK1 gene, which was previously pointed as a potentially oncogenic in breast cancer. This finding, on its own, underscore importance of this work and its potential for developing future clinical applications. There are several other interesting results that are supported by careful statistical analysis and balanced discussion.

I have several minor comments that are directed to making this manuscript and its Supplement more clear and useable to the broad cancer and genomics communities.

1. It may be useful to give a reference clarifying the term “adnexal” in relation to skin tumors. General search with the “adnexal” string leads overwhelmingly to gynecological tumors.
2. Authors use IntOGen (broad range functional predictions) and dNdScv (ratio between non-synonymous and synonymous changes) to detect potential driver genes. It will be helpful to add one more feature in describing mutations in these genes, i.e., the ratio between missense and null (nonsense and frameshifts) and to discuss functional implications. Comparing mutations in ALPK1 with mutations in CYLD (Fig. 2) it appears that the former has no stop-codon mutations, while in the latter they are present in abundance. Based on Suppl. Table 2 this also applies to presence or to the absence of frameshifts in these putative driver genes. Does this suggest that CYLD acts as tumor suppressor, while ALPK1 is an oncogene?
3. It would help others, who may be comparing mutation data from this study with TCGA data, if the lists of mutation calls (Suppl Table 2) would be organized similar to TCGA mutation annotation files (MAFs). In the TCGA MAFs, as well as in many other cancer genomics papers, sample IDs are presented in the single column, which simplifies partial extraction of the data after filtering as well as post-filtering counting.

4. Sequencing of morphologically benign precursor regions may provide important information about clonality in these areas as compared with cancers. Histograms of allele fractions of mutation calls could address this comparison.

5. Authors do compare mutation loads and base substitution spectra in adnexal tumors with TCGA tumors. It may be useful to compare also mutation signatures between these datasets. Of specific interest could be TCGA tumors with the same location (head and neck squamous cell carcinomas – HNSC). This could be limited to the major signatures in TCGA HNSC. For example, APOBEC signature detected in high grade tumors sequenced in in this study (Signatures 2 and 13, Fig. 3a) is also prominent in TCGA HNSC (PMID 23945592, PMID 23852170)

Reviewer #2 (Remarks to the Author):

The study identifies a novel hotspot mutation in ALKP1 in cutaneous spiradenoma and at lower frequency in histopathological related cylindromas. These are relatively rare adnexal tumors with sweat gland differentiation, that rarely can transform into carcinoma. The detailed and thorough study considerably expands our understanding of these tumors, which were previously only known to have loss of function mutations in CYLD. As shown here the ALKP1 mutations act in the same pathway, and similarly lead to activation of NFkB, which is shown in vitro and confirmed by immunohistochemistry in tumor samples. The authors also convincingly refute a prior claim that these tumors have MYB fusions, similar to adenoid cystic carcinomas, which is an important finding. Finally the authors show that transformation to carcinoma is accompanied by additional genetic alterations including TP53 and TERT promoter mutations, which provides diagnostic biomarkers for borderline cases. The conclusions of the study are consistent with the data and I think this will be an often cited study regarding the mutational landscape of this class of tumors,

My minor requests for clarification refer to the evolution from precursor lesions and the germline alterations.

1) The lack of overlap in somatic mutations between the benign regions and carcinomatous regions is difficult to follow/believe. How can it be that some of these lesions are not clonally related? The explanation that these adjacent neoplasms arise from different cells is not convincing. Could it be that shared somatic mutations were missed due to normal cell contamination in the less cellular areas? It would be good to expand on this and show corresponding photomicrographs.

2) The finding of PTEN and NBS1 germline mutations is intriguing but potentially misleading. What were the clinical phenotypes of these patients. Was there any evidence of tumor susceptibility syndrome (multiple primaries, early onset, other stigmata or genomic features associated with these germline mutations). What was the allelic status of the germline mutation in the tumor?

Boris Bastian

Reviewer #1:

Authors present the genomic analysis of Formalin-Fixed Paraffin-Embedded (FFPE) samples of skin adnexal tumors (cylindromas, spiradenomas, low- and high-grade spiradenocarcinoma as well as of morphologically benign precursor regions of these cancers. These are skin cancer with no or very little genomics information, so information presented by the study is important for a broad community of cancer researchers. The study indicated a specific recurrent somatic missense mutation in the kinase domain of ALPK1 gene, which was previously pointed as a potentially oncogenic in breast cancer. This finding, on its own, underscore importance of this work and its potential for developing future clinical applications. There are several other interesting results that are supported by careful statistical analysis and balanced discussion.

We thank Reviewer 1 for their kind words. We were excited to see the results of this study and the landscape of skin adnexal tumours.

I have several minor comments that are directed to making this manuscript and its Supplement more clear and useable to the broad cancer and genomics communities.

1. It may be useful to give a reference clarifying the term “adnexal” in relation to skin tumors. General search with the “adnexal” string leads overwhelmingly to gynecological tumors.

We have clarified this point in the first few lines of the introduction of the revised manuscript making clear that the focus of our study was skin adnexal tumours as distinct from gynecological adnexal tumors.

2. Authors use IntOGen (broad range functional predictions) and dNdScv (ratio between non-synonymous and synonymous changes) to detect potential driver genes. It will be helpful to add one more feature in describing mutations in these genes, i.e., the ratio between missense and null (nonsense and frameshifts) and to discuss functional implications. Comparing mutations in ALPK1 with mutations in CYLD (Fig. 2) it appears that the former has no stop-codon mutations, while in the latter they are present in abundance. Based on Suppl. Table 2 this also applies to

presence or to the absence of frameshifts in these putative driver genes. Does this suggest that *CYLD* acts as tumor suppressor, while *ALPK1* is an oncogene?

As suggested we have provided these numbers in the supplementary Table. Experimental work suggests that *CYLD* is a tumour suppressor [PMID:25342597] which is in keeping with the mutation profile we observe. Based on the functional overexpression studies in supplementary Figure 6 we suggest that *ALPK1* is functioning as an oncogene which makes sense given that the hotspot variant we identified is a missense mutation (V1092A).

3. It would help others, who may be comparing mutation data from this study with TCGA data, if the lists of mutation calls (Suppl Table 2) would be organized similar to TCGA mutation annotation files (MAFs). In the TCGA MAFs, as well as in many other cancer genomics papers, sample IDs are presented in the single column, which simplifies partial extraction of the data after filtering as well as post-filtering counting.

We have generated this file and include it in the supplementary data of the paper. We agree that it will help others work with the data we have generated as part of this study. We also provide the data in VCF, a format used by many investigators.

4. Sequencing of morphologically benign precursor regions may provide important information about clonality in these areas as compared with cancers. Histograms of allele fractions of mutation calls could address this comparison.

We reflected on this comment and the comment made by Prof. Bastian and decided to discuss just the *ALPK1* mutations in the benign precursor regions. The complexity of analysis of FFPE sequence data and the depth of coverage we generated may be insufficient to make an exome-wide comparison. We thank Reviewer 1 for this insightful comment.

5. Authors do compare mutation loads and base substitution spectra in adnexal tumors with TCGA tumors. It may be useful to compare also mutation signatures between these datasets. Of specific interest could be TCGA tumors with the same location (head and neck squamous cell carcinomas – HNSC). This could be limited to the major signatures in TCGA HNSC. For example, APOBEC signature detected in high grade tumors sequenced in in this study (Signatures 2 and 13, Fig. 3a) is also prominent in TCGA HNSC (PMID 23945592, PMID 23852170)

We thank Reviewer 1 for their constructive comments and have performed the analysis shown below. We obtained the somatic mutations from Alexandrov et.al. 2013 [PMID: 23945592]. We incorporated these data with the somatic mutations from our adnexal tumour cohort and computed the contribution of the 30 published mutational signatures (as described in the COSMIC database) in these samples using deconstructSigs. Each tumour sample was transformed in to a vector of length 31 where the first 30 values represent the contribution of the aforementioned signatures and the final value an unknown mutational process. This data matrix (which represents the contribution of published mutational signatures and one unknown signature) was then projected two-dimensionally using a t-SNE plot (Please see next page). As shown below there is no clustering of adnexal tumours

with any particular tumour type with the exception of several of the cylindromas clustering with melanomas, which appears to be due to Signature 7 (UV light mutations) in cylindromas from the head and neck as we discuss in our manuscript. Note: in the figure we have shown the head and neck cancers as (large) grey triangles and the adnexal tumours as (large) coloured circles. The other TCGA tumour types are shown as smaller triangles with the colour indicating the tumour type.

We thank Reviewer 1 for their constructive comments.

Reviewer #2:

The study identifies a novel hotspot mutation in ALKP1 in cutaneous spiradenoma and at lower frequency in histopathological related cylindromas. These are relatively rare adnexal tumors with sweat gland differentiation, that rarely can transform into carcinoma. The detailed and thorough study considerably expands our understanding of these tumors, which were previously only known to have loss of function mutations in CYLD. As shown here the ALKP1 mutations act in the same pathway, and similarly lead to activation of NFkB, which is shown in vitro and confirmed by immunohistochemistry in tumor samples. The authors also convincingly refute a prior claim that these tumors have MYB fusions, similar to adenoid cystic carcinomas, which is an important finding. Finally the authors show that transformation to carcinoma is accompanied by additional genetic alterations including TP53 and TERT promoter mutations, which provides diagnostic biomarkers for borderline cases. The conclusions of the study are consistent with the data and I think this will be an often cited study regarding the mutational landscape of this class of tumors,

We thank Prof. Bastian for reviewing our paper and for his comments below.

My minor requests for clarification refer to the evolution from precursor lesions and the germline alterations.

1) The lack of overlap in somatic mutations between the benign regions and carcinomatous regions is difficult to follow/believe. How can it be that some of these lesions are not clonally related? The explanation that these adjacent neoplasms arise from different cells is not convincing. Could it be that shared somatic mutations were missed due to normal cell contamination in the less cellular areas? It would be good to expand on this and show corresponding photomicrographs.

As above, we reflected on this comment and decided to discuss just the *ALPK1* mutations in the benign precursor regions. The complexity of the analysis of FFPE sequence data, the depth of coverage we generated and the possibility of contamination between compartments may make an exome-wide comparison problematic as noted by Prof. Bastian. We have altered our discussion accordingly.

2) The finding of PTEN and NBS1 germline mutations is intriguing but potentially misleading. What were the clinical phenotypes of these patients. Was there any evidence of tumor susceptibility syndrome (multiple primaries, early onset, other stigmata or genomic features associated with these germline mutations). What was the allelic status of the germline mutation in the tumor?

Prof. Bastian makes a very good point about the germline mutations we identified. The *NSD1* variant (R2017W) has been reported in ClinVar in three unrelated individuals with SOTO syndrome and is completely absent from all sequence variation databases (more than 250,000 normal individuals). It is defined by American College of Medical Genetics (ACMG) criteria as "pathogenic". Several additional

studies have also reported this variant in patients with SOTO syndrome [for example PMID:12807965] and it is predicted to disrupt the SET domain. Further, lymphoblastoid cell lines carrying the R2017W NSD1 mutation (OGS55) are used as cellular models of SOTO syndrome. The clinical records (those available to pathology) only describe the lesion that was excised and provide no morphometric information about the patient. *NSD1* is a haploinsufficient gene so we wouldn't expect to see somatic mutations in the WT allele [PMID:11896389].

The germline mutation we found in *PTEN* has been reported in a single individual in ClinVar with Cowden syndrome. This substitution occurs at a position that is conserved across species. In silico analysis predicts this variant is probably damaging to the protein structure/function. Missense (disease causing) mutations in nearby residues (V191G, M198K, T202I, M205V, S207R) have been reported, supporting the functional importance of this region of the protein. That said the clinical records for our patient do not mention Cowden syndrome and the patient was 60 when their skin tumour was removed. Further, we did not detect a mutation in the other allele of *PTEN*, which doesn't preclude its inactivation by other mechanisms, but does mean we should exercise caution. Thus, for both *NSD1* and *PTEN* we have tempered the discussion of these variant and made clear that further cases with disruptive alleles of these genes will need to be found to confirm their role in adnexal tumour predisposition.

We thank Prof. Bastian for his constructive comments and expert insights.

REVIEWERS' COMMENTS:

Reviewer #1 (Remarks to the Author):

Authors made significant changes and additions, which completely addressed reviewer's critique. I have no further suggestions to modify/supplement this submission.

Reviewer #2 (Remarks to the Author):

My comments have been addressed in this revised version, which has been further improved by the analyses and comments added in response to other reviewers.

REVIEWERS' COMMENTS:

Reviewer #1 (Remarks to the Author):

Authors made significant changes and additions, which completely addressed reviewer's critique. I have no further suggestions to modify/supplement this submission.

Reviewer #2 (Remarks to the Author): My comments have been addressed in this revised version, which has been further improved by the analyses and comments added in response to other reviewers.

We thank the reviewers for their helpful comments throughout the review process